# A lipoprotein/β-barrel complex monitors lipopolysaccharide integrity transducing information across the outer membrane

**Anna Konovalova, Angela M Mitchell, Thomas J Silhavy\***

Department of Molecular Biology, Princeton University, Lewis Thomas Laboratory, Princeton, United States

**Abstract** Lipoprotein RcsF is the OM component of the Rcs envelope stress response. RcsF exists in complexes with β-barrel proteins (OMPs) allowing it to adopt a transmembrane orientation with a lipidated N-terminal domain on the cell surface and a periplasmic C-terminal domain. Here we report that mutations that remove BamE or alter a residue in the RcsF trans-lumen domain specifically prevent assembly of the interlocked complexes without inactivating either RcsF or the OMP. Using these mutations we demonstrate that these RcsF/OMP complexes are required for sensing OM outer leaflet stress. Using mutations that alter the positively charged surface-exposed domain, we show that RcsF monitors lateral interactions between lipopolysaccharide (LPS) molecules. When these interactions are disrupted by cationic antimicrobial peptides, or by the loss of negatively charged phosphate groups on the LPS molecule, this information is transduced to the RcsF C-terminal signaling domain located in the periplasm to activate the stress response.

## Introduction

\*For correspondence: tsilhavy@ princeton.edu

**Competing interests:** The authors declare that no competing interests exist.

The outer membrane (OM) of Gram-negative bacteria is an asymmetric bilayer with lipopolysaccharide (LPS) and phospholipids in the outer and inner leaflets, respectively (*Silhavy et al., 2010*). LPS is a glycolipid that consists of three domains: lipid A, the core and the O-antigen (*Raetz and Whitfield, 2002*). Several sugars in lipid A and the core are phosphorylated conferring negative charge to the LPS molecule. In the OM, these negatively charged groups are bridged by divalent cations, which help to establish strong lateral interactions between LPS molecules. In addition to stabilizing the OM, these lateral interactions contribute to the unique barrier properties of the OM making it impermeable to hydrophobic compounds, detergents and dyes (*Nikaido, 2003*). The OM also protects Gram-negatives from the host innate immunity factors and antibiotics limiting their effectiveness. In order to disrupt the OM, many organisms produce cationic antibacterial peptides (CAMPs) that bind LPS (*Hancock and Diamond, 2000*). As a result of this binding, the OM is permeabilized and this not only facilitates further uptake of the CAMPs but also sensitizes Gram-negatives to antibiotics and host-factors, including lysozyme. For this reason, several CAMPs are "last hope" antibiotics against antibiotic-resistant Gram-negative bacteria (*Li et al., 2006*).

Because of the importance of OM integrity and barrier function for survival, Gram-negative bacteria have developed several envelope stress responses to monitor and combat environmental insults. One such envelope response, Rcs (Regulator of Capsule Synthesis), is activated strongly by OM and PG stress (*Majdalani and Gottesman, 2005*). Rcs controls the expression of capsule exopolysaccharides that are exported to the cell surface and help to stabilize the OM (*Gottesman et al., 1985*). In addition, Rcs downregulates flagella expression (*Francez-Charlot et al., 2003*), shifting bacteria from planktonic to a biofilm growth mode (*Ferrières and Clarke, 2003*; *Latasa et al., 2012*), which is often associated with further increased resistance. Rcs is conserved in *Enterobacteriaceae* and, for

**eLife digest** Many disease-causing bacteria have an outer membrane that surrounds and protects the cell, while many hosts of these bacteria produce molecules called antimicrobial peptides that disrupt this outer membrane. In response to this attack, bacteria have evolved a defense system to reinforce their membrane when antimicrobial peptides are present. However, it was not clear how the bacteria sensed these peptides.

One clue came from a recent discovery that the bacterial protein required for sensing the peptides is threaded through a barrel-shaped protein to expose a section of it on the bacterial cell's surface. Now, Konovalova et al. have tested if this surface-exposed domain directly detects damage to the outer membrane caused by the antimicrobial peptides.

The investigation revealed several mutants of *Escherichia coli* that still make the sensor protein but are unable to thread it through the barrel-shaped protein and place a portion on the cell surface. Konovalova et al. showed that these mutants are essentially "blind" to the presence of antimicrobial peptides, and thus prove that it is the surface-exposed domain that works as the sensor. Antimicrobial peptides bind to a major component of the outer membrane and disrupt its normal interactions. Further experiments showed that positively charged sites in surface-exposed domain of the sensor are required to detect these changes and transmit this information inside the cell.

Future studies are now needed to understand how the sensor is assembled inside the barrel-shaped protein, and how the danger signal is sent across the membranes that envelope bacterial cells to activate the defense system inside the cell.

many enteric pathogens, it is important for virulence and/or survival in the host (*Erickson and Detweiler, 2006*; *Hinchliffe et al., 2008*).

Rcs is one of the most complex signal transduction pathways in bacteria involving at least seven proteins in four different cellular compartments. RcsF is an OM lipoprotein that acts as a sensory component (*Majdalani and Gottesman, 2005*). Unlike most lipoproteins in *E. coli*, RcsF is anchored to the outer leaflet of the OM by its lipid moiety and contains an N-terminal domain that is surface exposed (*Konovalova et al., 2014*). The short, hydrophilic transmembrane domain of RcsF is threaded through the lumen of β-barrel proteins (OMPs) exposing the C-terminal signaling domain in the periplasm (*Konovalova et al., 2014*). The inner membrane (IM) protein RcsC is a hybrid histidine kinase, which autophosphorylates and passes the phosphate through the IM phosphotransferase protein RcsD to a cytoplasmic DNA-binding response regulator RcsB (*Stout and Gottesman, 1990*; *Takeda et al., 2001*; *Majdalani and Gottesman, 2007*). RcsB, either alone or in combination with other regulators, such as RcsA (*Stout et al., 1991*), BglJ (*Venkatesh et al., 2010*) or GadE (*Krin et al., 2010*) regulates expression of target genes. In addition, IM protein IgaA negatively regulates the activity of RcsC (*Cano et al., 2002*; *Domínguez-Bernal et al., 2004*). Based on genetic analysis as well as some interaction studies it has been proposed that RcsF can interact directly with the periplasmic domain of IgaA to alleviate inhibition of RcsC (*Domínguez-Bernal et al., 2004*; *Cho et al., 2014*).

Although the signal transduction pathway itself is reasonably well characterized, how RcsF senses the many different envelope defects that induce this system remain poorly understood. The fact that LPS is found exclusively in the outer leaflet of the OM and that RcsF is required to sense LPS structural defects prompted us to search for a surface-exposed domain. Our discovery that RcsF exists in a transmembrane complex with OMPs suggested that this interlocked structure is required for sensing the LPS defects (*Konovalova et al., 2014*). However, another study, which focused primarily on Rcs activated by PG stress caused by A22 or mecillinam treatment, proposed a model in which the RcsF/OMP complex has no function in RcsF sensing and signaling and only serves an inhibitory role during steady state growth. This model was based on the observation that Rcs induction by A22 depends on newly synthesized RcsF (*Cho et al., 2014*). Here, we show that the RcsF/OMP complex is functional and that LPS defects are sensed directly by the surface-exposed domain of RcsF.

## Results

### Low levels of polymyxin B cause a specific OM defect

Several mutations in the LPS biosynthesis pathway that result in the production of LPS molecules with altered structure are known to activate Rcs (*Parker et al., 1992*). However, the use of chemical inducers avoids phenotypic adaptation and enables kinetic analysis, which can provide important insights. Therefore, we sought a small molecule that could be used to test the function of RcsF/OMP complexes in sensing LPS perturbations.

Polymyxin B (PMB) is a CAMP with bactericidal activity and its mechanism of action is well established (*Daugelavicius et al., 2000*). When used at the minimal inhibitory concentration (MIC, 2–8 µg/ml), PMB binds to LPS and destabilizes the OM outer leaflet resulting in marked increase in OM permeability. When cells are incubated with much higher concentrations of PMB, it can also integrate into the IM causing a lethal disruption of the membrane potential. PMB is a known RcsF-dependent inducer of the Rcs pathway (*Farris et al., 2010*).

In the following experiments, we used 0.5 µg /ml PMB to treat cells in mid-log phase (OD600 of 0.5, $5*10^8$ cells/ml). At this cell density, the MIC of PMB was determined to be 8 µg /ml. Therefore, the concentration of PMB we used to induce the Rcs system is more than 10 fold below the MIC value and growth is not affected under these conditions.

We confirmed that 0.5 µg/ml PMB causes OM, but not IM, stress in our strain background using two routine assays developed to study the effect of CAMPs on the bacterial envelope (*Figure 1A and B*) (*Loh et al., 1984*; *Wu et al., 1999*). The NPN (1-N-phenylnaphthylamine) uptake assay is used to quantitatively monitor OM permeability caused by CAMPs (*Loh et al., 1984*). NPN is an environmentally sensitive fluorescent dye that fluoresces in the membrane environment but not in solution. Due to its hydrophobic nature, it cannot penetrate Gram-negative OM due to the presence of LPS. However, when CAMPs compromise the OM, NPN enters the cell and integrates into membranes resulting in an increase in fluorescence. When mid-log cells were incubated with 0.5 µg /ml PMB we observed only a small increase in fluorescence, about 1.5 fold, compared to a 6-fold increase when cell were treated with 8 µg /ml PMB (*Figure 1A*).

The diSC3(5) (Dipropylthiadicarbocyanine iodide) release assay is based on a membrane potential-sensitive fluorescent dye (*Wu et al., 1999*). DiSC3(5) accumulates in the IM in a proton motive force (PMF)-dependent manner, where it self-quenches the fluorescence. However, when PMF is compromised, diSC3(5) is released from the IM resulting in increased fluorescence. We treated diSC3(5) -labeled cells with 0.5 µg /ml PMB or an MIC (12.5 µg /ml) of Gramicidin, a CAMP known to be PMF uncoupler. Gramicidin A but not PMB caused an increased fluorescence due to diSC3(5) release (*Figure 1B*). Taken together, these assays demonstrate that at low concentrations PMB does not cause IM depolarization and generates only a small increase in OM permeability, consistent with previous results (*Daugelavicius et al., 2000*)

We monitored the induction kinetics of the Rcs system in response to PMB using a chromosomal P*rprA-lacZ* reporter fusion (*Majdalani et al., 2002*). *rprA* encodes a small regulatory RNA that stimulates translation of the mRNA for the stationary phase σ factor RpoS (*Majdalani et al., 2002*). Expression of *rprA* is regulated exclusive by RcsB and P*rprA-lacZ* is used as a specific reporter for Rcs stress response activation (*Majdalani and Gottesman, 2007*). For this experiment, we grew the MC4100 *PrprA-lacZ* strain (from now on WT, the parent for all strains) to midlog phase. We then added PMB and followed the Rcs induction over time by monitoring expression of P*rprA-lacZ* reporter using qRT-PCR and β-galactosidase assays. PMB causes a strong and almost immediate induction of Rcs, as quickly as 2 min on the RNA level and 10 min on the protein level (*Figure 1C*).

In order to test whether PMB induces other envelope stress responses or is specific for Rcs, we followed the activity of two other major envelope stress responses, Sigma E and Cpx by monitoring the expression of well-established markers, *rpoE* (*Mutalik et al., 2009*) and *cpxP* (*Danese and Silhavy, 1998*). *Figure 1D* shows using qRT-PCR that PMB did not induce either of these stress responses, demonstrating that PMB induces only Rcs.

We also monitored OMP levels during PMB treatment to see whether PMB causes OMP assembly defects. Levels of the three major OMPs did not change upon PMB treatment (*Figure 1E*). Because we did not observe OMP assembly defects and because the Sigma E response, a sensitive monitor of OMP assembly, was not induced we conclude that low levels of PMB do not inhibit the Bam complex.

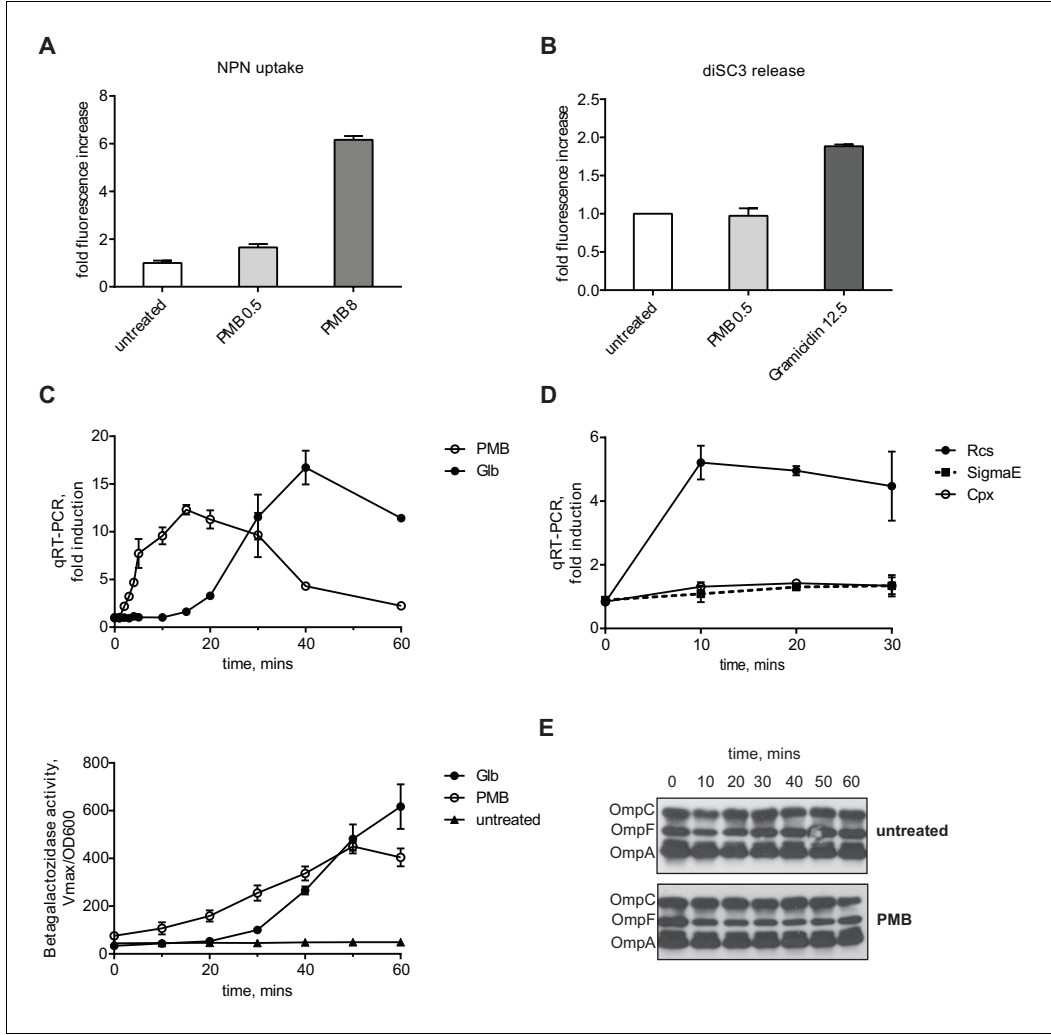

**Figure 1.** PMB causes a specific OM defect. (**A**) PMB at 0.5 μg/ml causes a slight OM permeability defect based on increased uptake and fluorescence of NPN dye. Graphs represent mean normalized end-point fluorescence +/- SD, n=3. (**B**) This concentration of PMB does not cause depolarization of the IM. Unlike gramicidin, PMB is unable to release PMF-dependent DiSC3(5) dye. Graphs represent mean normalized end-point fluorescence +/-SD, n=3. (**C**) Kinetics of Rcs induction upon PMB and Glb treatment at the mRNA (upper panel) and protein level (lower panel). Induction was monitored using a chromosomal P*rprA-lacZ* reporter by qRT-PCR or β-galactosidase assays. For mRNA quantification, graphs represent relative expression values normalized to a no treatment control for each time point +/- SD. β-galactosidase activity represent mean Vmax normalized to OD600, +/- SEM, n=3. (**D**) PMB induces Rcs but not the Cpx or SigmaE stress responses. Induction was monitored by following the relative expression of P*rprA-lacZ* (Rcs), *cpxP* (Cpx) or *rpoE* (SigmaE) by qRT-PCR. Graphs represent mean +/- SEM, n=3. (**E**) PMB does not cause OMP assembly defects based on immunoblot analysis.

The following figure supplement is available for figure 1:

**Figure supplement 1.** Glb induces Rcs but not the SigmaE stress response.

Taken together, we conclude that at these sub-MIC levels, PMB causes a specific OM defect.

## The PMB-induced Rcs response is independent of *de novo* protein synthesis

Globomycin (Glb) is an inhibitor of LspA, the lipoprotein signal peptidase (*Inukai et al., 1978*; *Yamagata et al., 1983*; *Dev et al., 1985*). Glb prevents maturation of lipoproteins resulting in the

accumulation of OM lipoproteins in the outer leaflet of the IM. It is well established that, when RcsF accumulates in the outer leaflet of the IM, due to Lol-avoidance mutations that alter the signal sequence or defects in lipoprotein maturation and/or export, it strongly induces the Rcs response (*Shiba et al., 2004*; *2012*; *Tao et al., 2012*). Indeed, we observed strong Rcs induction in response to 5 µM Glb (0.5 MIC). This induction required at least 15 min on the RNA level and 20 min on the protein level (*Figure 1C*). Clearly, the kinetics of Rcs induction are much slower with Glb than with PMB. Glb will prevent assembly of essential lipoproteins involved in both LPS and OMP assembly; however, depletion of these proteins to levels low enough to interfere with these assembly processes requires generations (*Malinverni et al., 2006*; *Wu et al., 2006*). Indeed, Glb does not induce the Sigma E response even after 60 min exposure (*Figure 1—figure supplement 1*). Therefore, we conclude that Glb is a chemical inducer of Rcs that is independent of OM damage.

There is no reciprocal transport of lipoproteins from the OM to the IM; therefore, only newly synthesized OM lipoproteins accumulate in the IM under Glb treatment. Therefore, the kinetics of the Glb-induced Rcs response can also serve as a temporal marker for induction due to mislocalization of newly synthesized RcsF. The observation that Glb induced Rcs with slower kinetics than with PMB suggested that induction with PMB does not require newly synthesized RcsF. To test this directly, we used the antibiotic Kasugamycin (Ksg), which is an inhibitor of translation initiation (*Okuyama et al., 1971*). After 15 min pre-treatment with Ksg we added PMB or Glb and followed the expression of P*rprA-lacZ* by qRT-PCR (*Figure 2*). As expected Ksg completely abolished Rcs induction in response to Glb (*Figure 2*, lower panel), however Ksg did not block the Rcs response to PMB, demonstrating that Rcs induction is independent of *de novo* protein synthesis and does not require newly synthesized RcsF (*Figure 2*, upper panel). Since RcsF is transported to the OM immediately after it is translocated from the cytoplasm, this result suggests that RcsF molecules already located in the OM are required to respond to PMB treatment.

## RcsF/OMP complexes are required for sensing OM stress

RcsF forms an interlocked complex with multiple OMPs, such as OmpA, OmpC and OmpF (*Konovalova et al., 2014*). However, it is unclear how much uncomplexed RcsF is present in the OM. Since OmpA is the major RcsF-interacting partner (*Konovalova et al., 2014*), we reasoned that loss of OmpA will lead to significant reduction in RcsF/OMP complexes (*Figure 3A and B*) and, if this complex senses LPS defects, this reduction should affect the response to PMB but not to Glb.

We followed the kinetics of Rcs induction in response to PMB and Glb in the WT and the *ompA* mutant by monitoring normalized LacZ activity produced by the P*rprA-lacZ* reporter over time (*Figure 3C*). PMB treatment did not induce Rcs in the *ompA* mutant (Figure. 3C, upper panel). In contrast, Rcs was induced in the *ompA* mutant upon Glb treatment with similar kinetics to that observed for the WT (*Figure 3C*, lower panel) suggesting that *ompA* is required specifically for PMB-induced OM stress.

OmpA is one of the most abundant OMPs in *E. coli*. To confirm that lack of Rcs response to PMB is the result of loss of RcsF/OMP complexes and not due to damage caused by the loss of the OmpA protein itself, we sought to identify RcsF/OMP assembly defective mutants that would lack RcsF/OmpA complexes but express both the individual proteins. Here we describe two such mutants (*Figure 3A and B*).

RcsF residue A55 is within the region of RcsF predicted to reside in the lumen of OMPs based on the results of site-specific photo-crosslinking (*Figure 3A*) (*Konovalova et al., 2014*). In the course of constructing pBPA variants for site-specific crosslinking, we noticed that the A55-pBPA variant was not functional for LPS sensing and did not crosslink to OMPs. We hypothesized that this residue might be important for RcsF/OMP assembly. Here we report that a mutation *rcsF_A55Y* results in a significant reduction in the assembly of RcsF/OmpA complexes (*Figure 3B*, upper panel). Photo-crosslinking also shows that RcsF interacts with BamA and it is thought that this reflects a role for BamA in the assembly of the RcsF/OMP complexes (*Konovalova et al., 2014*). The *rcsF_A55Y* mutation also results in a significant reduction in the levels of RcsF/BamA complex.

*bamE* encodes one of the non-essential lipoproteins of the Bam complex (*Sklar et al., 2007*). *bamE* mutants displayed a striking phenotype in RcsF assembly: levels of RcsF/OmpA complexes are severely reduced (*Figure 3B*, upper panel) and increased RcsF/BamA crosslinking is observed (*Figure 3B*, upper panel). Levels of the individual proteins, RcsF, OmpA or BamA, were not affected in these assembly mutants (*Figure 3B*, lower panel).

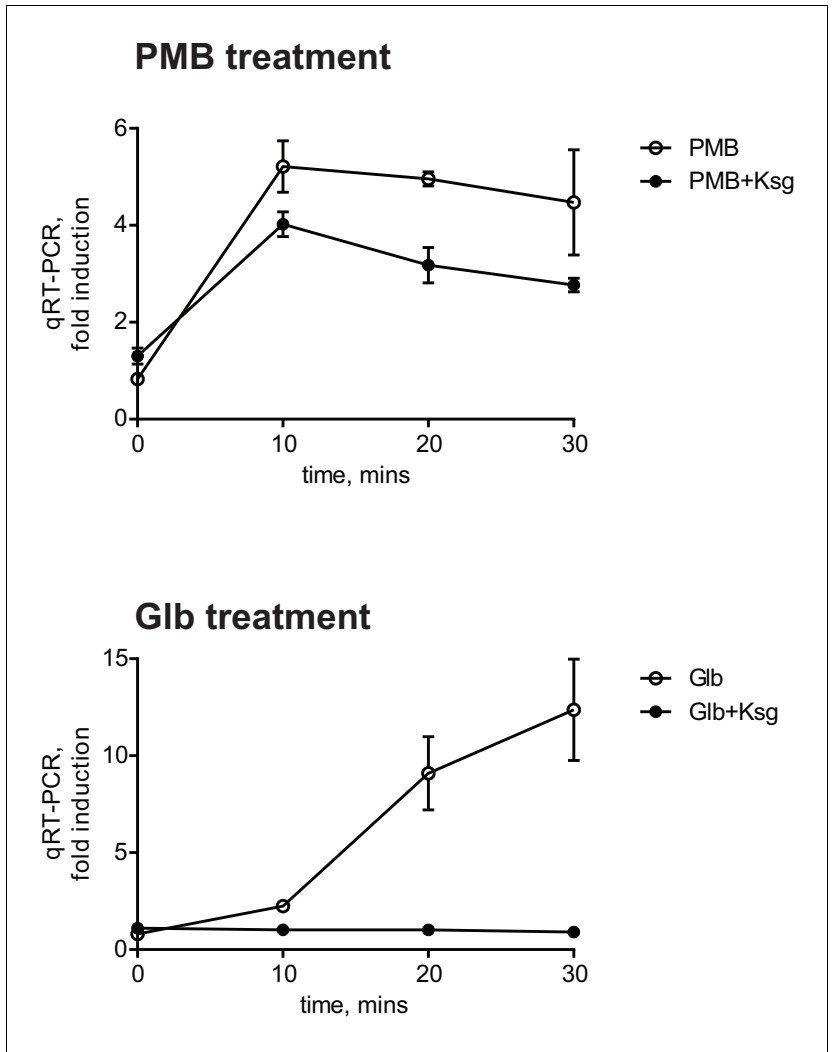

**Figure 2.** The PMB-induced Rcs response is independent of *de novo* protein synthesis. Cell cultures were pretreated with Ksg to inhibit protein synthesis for 15 min prior to addition of PMB or Glb. Rcs induction was then monitored by qRT-PCR. Graphs represent relative expression values normalized to a no treatment control for each time point, mean +/- SEM, n=3

Both of the mutants, *rscF_A55Y* and *bamE*, failed to activate Rcs in response to PMB induction (*Figure 3C*, upper panel); however, both responded to Glb treatment (*Figure 3C*, lower panel). LacZ activity was lower than the WT, likely due to the lower steady state levels of LacZ activity in the untreated *rscF_A55Y and bamE* cells.

To verify that RcsF/OMP complexes are more generally required for sensing LPS-stress and not specific to PMB, we tested the effect of the *ompA, rcsF_A55Y* and *bamE* mutations on Rcs signaling in the LPS biosynthesis mutant *waaP (rfaP)* (*Figure 3D*). *waaP* encodes lipopolysaccharide core heptose one (Hep(I)) kinase which catalyzes the addition of a negatively charged phosphate group to the LPS core (*Parker et al., 1992*; *Yethon et al., 1998*). *waaP* null mutations induce Rcs approximately 8 fold over the WT, resulting in a mucoid phenotype (*Figure 3—figure supplement 2*). Deletion of *ompA* in *waaP* background resulted in a strong synthetic interaction and this strain had a growth defect in liquid culture (*Figure 3—figure supplement 2*). For this reason, we did not perform experiments with this strain. When the *rcsF_A55Y* or *bamE* mutation was introduced in the *waaP* strain, no growth defect was observed demonstrating that the loss of OmpA not the RcsF/OmpA complex

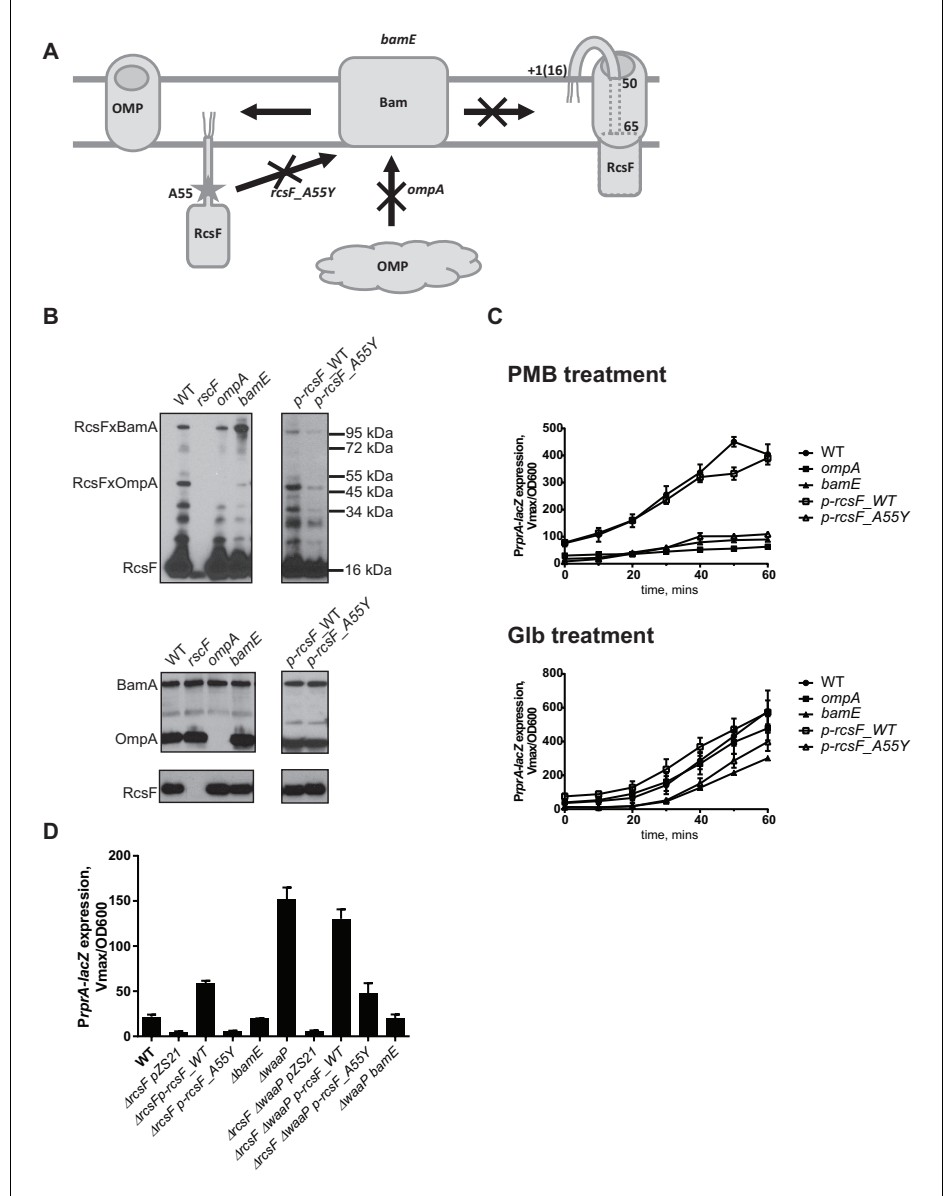

**Figure 3.** RcsF/OMP complexes are required for sensing OM stress. (**A**) Topology and assembly pathway of RcsF/OMP complexes (based on (14)). The lipidated N-terminus of RcsF is anchored in the outer leaflet of the OM exposing residues 16–49 on the cell surface. The transmembrane segment (residues 50–65) of RcsF is threaded through the lumen of the OMP exposing the C-terminal domain in the periplasm. RcsF/OMP complexes are assembled by the Bam machine. Not all OMPs are complexed with RcsF. The effect of different mutants used in this study on the assembly of RcsF/OMP complexes is shown. (**B**) The effect of *ompA, bamE* and *rcsF_A55*Y mutations on RcsF crosslinking to BamA and OmpA (upper panel) and the total RcsF, BamA and OmpA levels based on immunoblot analysis. (**C**) The *ompA, bamE* and *rcsF_A55*Y mutants do not respond to PMB (upper panel) but respond to Glb (lower panel) treatment based on expression of P*rprA-lacZ*. Graphs represent mean β-galactosidase activity +/- SEM, n=3 (**D**) The *bamE* and *rcsF_A55*Y mutants result in decreased P*rprA-lacZ*. Graphs represent mean β-galactosidase activity +/- SEM, n=3. Corresponding OD600 graphs and untreated controls are shown in *Figure 3—figure supplement 1*.

The following figure supplements are available for figure 3:

**Figure supplement 1.** Growth and kinetcs of Rcs induction in the assembly mutants including untreated cotrols.

**Figure supplement 2.** Growth phenotype of the assembly mutants.

causes this synthetic interaction (*Figure 3—figure supplement 2*). Strikingly, both assembly mutations significantly reduced the expression of the P*rprA-lacZ* reporter in the *waaP* strain and resulted in loss of the mucoid phenotype (*Figure 3D*, *Figure 3—figure supplement 2*) indicating that Rcs was not induced. P*rprA-lacZ* expression in the *rcsF_A55Y* strain was still somewhat stimulated by the *waaP* mutation, likely because the assembly of RcsF/OMP complexes was not completely abolished. Taken together, we conclude that RcsF/OMP complexes are required for sensing LPS defects.

## Positive charge of the surface-exposed region of RcsF is required for LPS-sensing

Mutations in several LPS biosynthesis genes result in Rcs induction (*Parker et al., 1992*). We systematically analyzed the effect of mutations in non-essential genes in the LPS biosynthesis pathway for the ability to induce Rcs (*Figure 4A,B*). We found mutations that result in defects in inner core biosynthesis and phosphate modification strongly induce Rcs (*Figure 4B*). *waaCFPG* mutations confer a phenotype, known as a deep rough phenotype; these strains are mucoid and unlike the WT are sensitive to detergents, bile salts and hydrophobic antibiotics (*Austin et al., 1990*; *Kamio and Nikaido, 1976*; *Parker et al., 1992*). To differentiate between altered LPS structure and increased permeability as potential inducing signals for Rcs we analyzed several OM biogenesis mutants that also display strong sensitivities to antibiotics without affecting LPS structure.

First, we tested the double *mlaA pldA* mutant, which knocks out two complementary pathways controlling OM lipid asymmetry (*Dekker, 2000*; *Malinverni and Silhavy, 2009*). This mutant has substantially increased levels of phospholipids in the outer leaflet of the OM (*Malinverni and Silhavy, 2009*). *Figure 4C* shows that Rcs is not induced in this strain. In addition, we tested mutants in the LPS export pathway, *lptD4213* and *lptE_R91D,K136D* (*Ruiz et al., 2005*; *Malojčić et al., 2014*). Rcs was also not induced in these strains (*Figure 4C*). The permeability phenotype of *lptE_R91D,K136D* is mild but *mlaA pldA* and *lptD4213* are as sensitive to detergents and hydrophobic antibiotics as deep rough mutants, e.g. *waaP* (*Figure 4D*). Importantly, RcsF was not generally inhibited in these strains, because it could be activated by introducing the *waaP* mutation (*Figure 4C*). These results demonstrate that OM permeability and/or disrupted asymmetry of the OM is not a physiological signal for Rcs.

The result above suggested that RcsF senses alterations of LPS structure in *waa* mutants rather than permeability. Mutations in *waaC, waaF* and *waaG* result not only in a truncated core (*Figure 4B*), but also in the loss of core phosphorylation (*Yethon et al., 2000*; *Yethon and Whitfield, 2001*) because the complete inner core with the first glucose is a substrate for WaaP kinase (*Yethon et al., 2000*; *Yethon and Whitfield, 2001*). The *waaP* mutation does not introduce core truncations but results in the loss of a majority of these core phosphates because WaaP activity is a prerequisite for WaaY phosphorylation of Hep(II) (*Yethon et al., 1998*). Our results demonstrate that *waaP* mutation and therefore decreased phosphorylation of LPS is sufficient to fully induce Rcs (*Figure 4B*).

LPS phosphates are essential for establishing cation-mediated LPS cross-bridges in the OM. LB is a $Mg^{2+}$-limiting medium, and does not contain a sufficient amount of cations to saturate LPS (*Papp-Wallace and Maguire, 2008*; *Nikaido, 2009*). Addition of $Mg^{2+}$ is known to stabilize the OM of the deep rough mutants (*Chatterjee et al., 1976*), likely through stabilization of the lipid A phosphates. Interestingly, when 10 mM $Mg^{2+}$ was added to LB, it decreased Rcs induction in *waaP* mutant (*Figure 4E*).

The two-component PhoPQ system responds to low $Mg^{2+}$ concentrations (*Soncini et al., 1996*; *Kato et al., 2003*). PhoPQ and Rcs are known to have partially overlapping regulons in several enterobacteria (*Hagiwara et al., 2003*; *García-Calderón et al., 2007*). Unlike in Salmonella, PhoPQ in *E. coli* does not regulate lipid A modifications through the Pmr pathway (*Winfield and Groisman, 2004*; *Rubin et al., 2015*). Because we observed a $Mg^{2+}$ dependent effect on Rcs in a *waaP* mutant, we analyzed Rcs induction in *waaP phoP* mutant (*Figure 4E*). Rcs was still induced and responded to $Mg^{2+}$-addition in the *waaP phoP* mutant (*Figure 4E*). Therefore, we concluded that $Mg^{2+}$ dependent Rcs signaling in *waaP* mutants was independent of PhoPQ and is likely a result of reinforced cross-bridges between lipid A-phosphates.

The effect of LPS charge and presence of cations on Rcs signaling lead us to suggest that RcsF senses the strength of LPS lateral interactions. Binding of a cationic peptide, such as PMB to LPS would also neutralize the charge and weaken LPS lateral interactions. PMB also contains lipid in

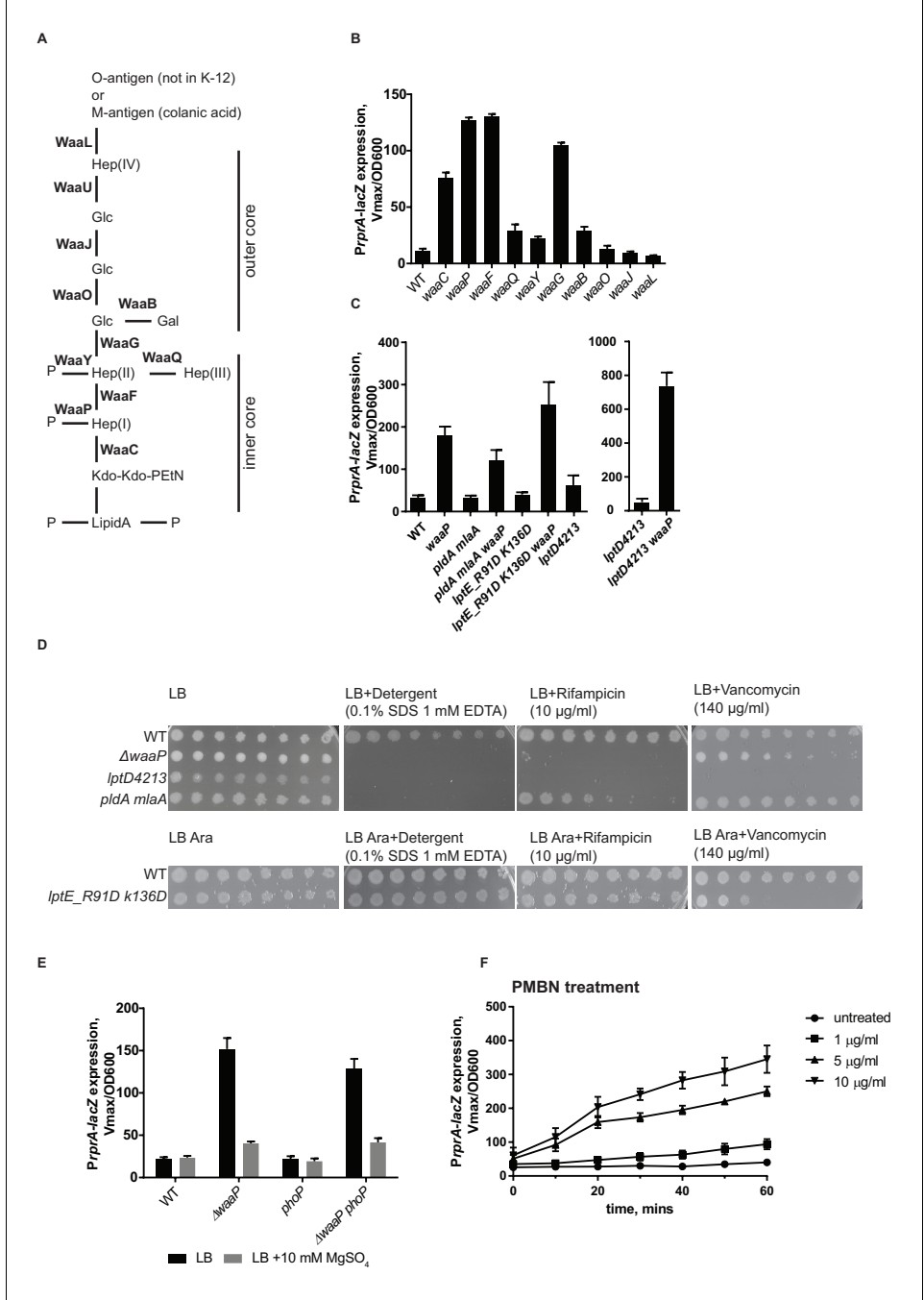

**Figure 4.** RcsF senses alteration in LPS structure. (**A**) Structure and biosynthesis of *E. coli* K-12 LPS according to (**Raetz and Whitfield, 2002**). Non-essential enzymes are shown in bold. (**B**) The effect of mutations in non-essential genes in the LPS biosynthesis pathway on Rcs induction based on P*rprA-lacZ* expression. (**C and D**) OM permeability is not a physiological inducing signal for Rcs. (**C**) Mutations that cause defects in asymmetry (*pldA mlaA*) or LPS export (*lptD4213 and lptE_R91D K136D*) do not induce Rcs. RcsF is not generally inhibited in these strains because Rcs can still be induced by introducing the *waaP* mutation. (**D**) Mutations *pldA mlaA, lptD4213 and lptE_R91D K136D* confer OM permeability defects (see text for references) assayed by plating 10-fold serial dilutions of overnight cultures onto LB plates supplemented with antibiotics or detergents. Note, arabinose (Ara) is required for growth of *lptE_R91D K136D* mutant. (**E**) The addition of $Mg^{2+}$ reduces Rcs signaling in the *waaP* background in a *phoP*-independent manner. (**F**) Lipid-truncated PMB derivative, PMBN, induces Rcs in concentration-dependent manner. Graphs B, C, E and F represent mean β-galactosidase activity +/- SEM, n=3

addition to a cyclic peptide ring, and the lipid integrates into and can disorganize the membrane bilayer (*Vaara, 1992*). We therefore tested whether charge-neutralization of LPS by PMB is sufficient to induce Rcs. For this, we analyzed the ability of a lipid-truncated PMB derivative, PMB nonapeptide (PMBN) to induce Rcs. PMBN also neutralizes LPS but does not disturb the bilayer and is non-toxic. Like PMB, PMBN also induces Rcs, but higher concentrations are required to offset the lower affinity of binding to LPS (*Figure 4F*) (*Vaara and Viljanen, 1985*; *Thomas and Surolia, 1999*). Therefore, we conclude that charge-neutralization of LPS by PMB is sufficient to induce Rcs.

If RcsF/OMP complexes sense LPS defects directly, it seems likely that the surface-exposed domain of RcsF acts as the sensory domain. One of the interesting features of surface-exposed

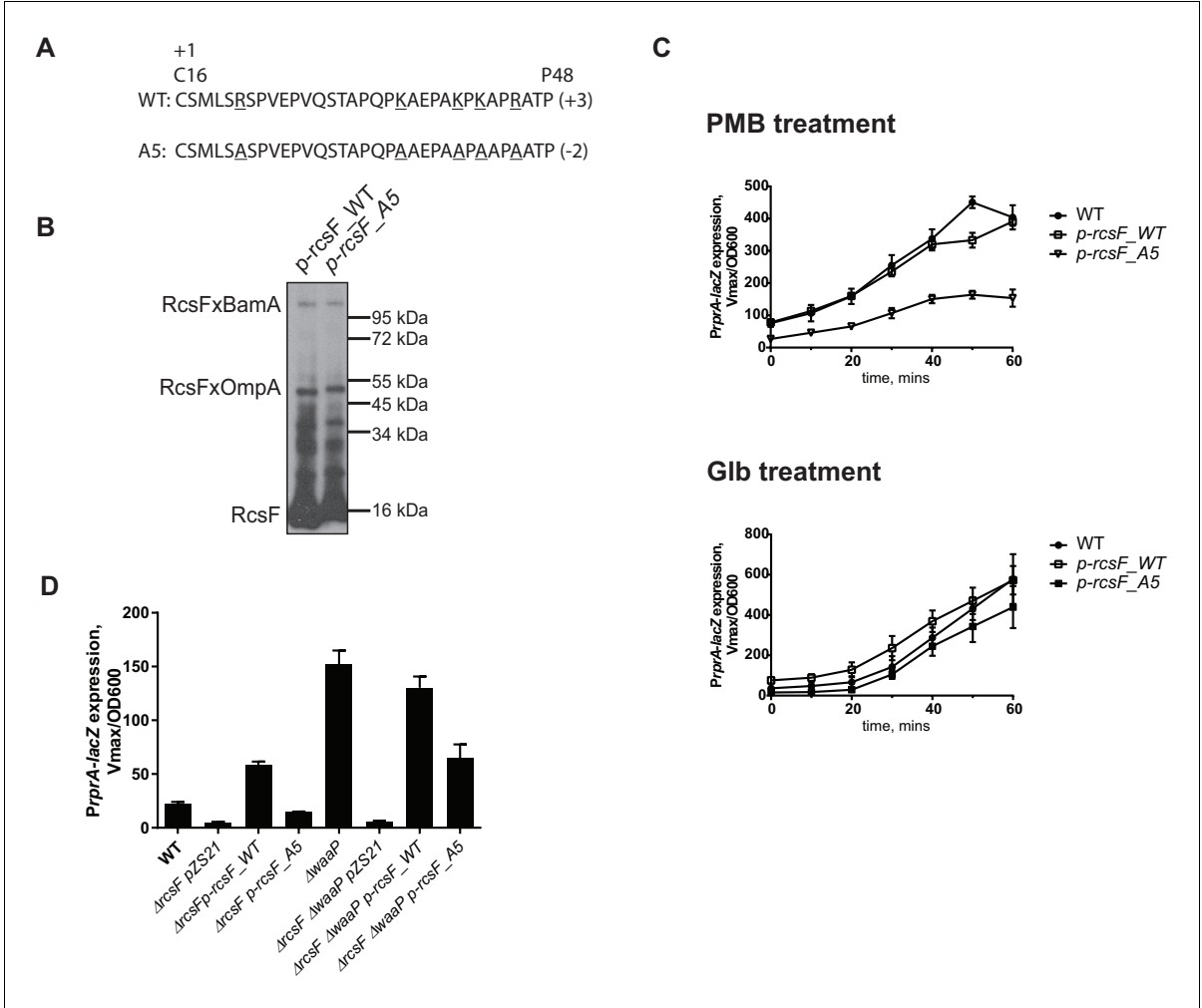

**Figure 5.** Positive charge of the surface-exposed region of RcsF is required for LPS-sensing. (**A**) Amino acid sequences and total charge of the surface exposed domain of RcsF_WT and the charge substitution mutant RcsF_A5. Positively charged residues (Lys and Arg) which were substituted by Ala are underlined. (**B**) Charge substitution mutations do not affect RcsF crosslinking to BamA and OmpA. (**C**) The *rcsF_A5* mutant does not respond to PMB (upper panel) but does respond to Glb (lower panel) treatment based on expression of a P*rprA-lacZ* reporter analyzed by beta-galactosidase assay. (**D**) The *rcsF_A5* mutant results in decreased P*rprA-lacZ* expression in the *waaP* background. Graphs C and D represent mean β-galactosidase activity +/- SEM, n=3. Corresponding OD600 graphs and untreated controls are shown in *Figure 5—figure supplement 1*.

The following figure supplements are available for figure 5:

**Figure supplement 1.** Kinetics of P*rprA-lacZ* expression in RcsF/OMP charge substitution strains upon treatment with PMB, Glb (same as *Figure 5C*) and untreated controls together with corresponding growth curves.

**Figure supplement 2.** Plate phenotype of the charge substitution mutant in the WT and *waaP* background.

region of RcsF is the abundance of positively charged amino acids (*Figure 5A*). To investigate the role of these positive charges in sensing LPS defects, we substituted the five Arg and Lys residues in the surface-exposed region (16–48) with Ala to generate a charge substitution mutant gene we refer to as *rcsF_A5* (*Figure 5A*). We verified that RcsF/OmpA complexes are formed with the same efficiency as in RcsF_WT (*Figure 5B*) and that this mutant protein responded normally to Glb (*Figure 5C*, lower panel). Next, we tested how this mutant protein responded to PMB (*Figure 5C*, upper panel). Although not completely abolished, the Rcs response to PMB was significantly reduced as compared to WT (*Figure 5C*, upper panel). Likewise, introduction of *rcsF_A5* into the *waaP* background results in significantly reduced expression of P*rprA-lacZ* and loss of the mucoid phenotype (*Figure 5D*, *Figure 5—figure supplement 2*). We conclude that the positive charge of surface-exposed region of RcsF is important for sensing alterations in LPS.

## Discussion

The lipidated amino terminus of lipoprotein RcsF is displayed on the cell surface; the carboxy-terminal signaling domain resides in the periplasm and the short, hydrophilic transmembrane domain is protected from the lipid environment by threading through an OMP, most often OmpA. We show here that this interlocked, heterodimeric structure functions directly to sense known Rcs inducing signals that cause perturbations in the outer leaflet of the OM caused either by treatment with CAMPs or by mutations that cause LPS structural defects. We show that altered OM permeability or asymmetry are not physiological inducers of RcsF. Instead, RcsF senses the state of LPS lateral interactions and it is activated when these interactions are perturbed either by: i) neutralization of LPS negative charge by CAMPs (such as PMB) as a result of direct binding; ii) decreased LPS phosphorylation as a result of mutations in LPS biosynthesis pathway; iii) by lack of cations to stabilize LPS cross-bridges. Moreover, we show that the positive charge of surface-exposed domain of RcsF is necessary for sensing these LPS defects.

An alternative model for RcsF signaling has been proposed by *Cho et al. (2014)*. This model proposes that in unstressed cells, RcsF is constantly assembled into RcsF/OmpA complexes that have no function in sensing and signal transduction. Signals that induce Rcs do so by inhibiting the function of the Bam complex. This prevents RcsF/OMP assembly and allows newly synthesized RcsF to remain free in the periplasm where it can engage the IM components of the signal transduction system. According to this model, activation of Rcs signaling depends on *de novo* protein synthesis (*Cho et al., 2014*).

Here, we have provided multiple lines of evidence demonstrating that this model cannot explain how RcsF senses changes in the OM outer leaflet. First, as noted above, we show that the surface-exposed domain of RcsF/OmpA functions directly to sense these defects. Second, we provide evidence that PMB does not affect β-barrel assembly nor does it induce the SigmaE stress response, which is a sensitive indicator of Bam complex function and the presence of unfolded OMPs in the periplasm. Finally, we show that PMB induction of the Rcs system happens even in the presence of a protein synthesis inhibitor.

Two other categories of mutations or chemical agents are also known Rcs inducers: i) perturbations affecting lipoprotein maturation, such as *pgsA* (*Shiba et al., 2004*), Glb (this study), or trafficking (*lolA* depletion) (*Tao et al., 2012*); ii) perturbations in PG biogenesis, such as those caused by treatment with mecillinam and other β-lactams (*Laubacher and Ades, 2008*), lysozyme (*Callewaert et al., 2009*; *Ranjit and Young, 2013*), A22 (*Cho et al., 2014*), or genetically by introducing multiple *pbp* knockouts (*pbp4, 5, 7, ampH*) (*Evans et al., 2013*).

It is well established that perturbations in lipoprotein assembly lead to the accumulation of RcsF in the inner membrane allowing interaction with downstream components (*Shiba et al., 2012*). Using Ksg, we show that this mechanism clearly depends on RcsF *de novo* synthesis and is independent of RcsF/OMP assembly in the OM.

Activation of RcsF in response to perturbations in PG biogenesis also depends on protein synthesis and is likely independent of RcsF/OMP complexes (*Cho et al., 2014*). However, it is unclear how PG defects induce Rcs. RcsF does not interact with PG, and therefore, activation might not be direct. For example, β-lactams (including mecillinam), A22 and mutations in *pbp* genes induce not only Rcs but also the Cpx response (*Laubacher and Ades, 2008*; *Delhaye et al., 2016*; *Evans et al., 2013*) and, at least in the case of the *pbp4, 5, 7, and ampH* mutations, Rcs induction relies fully on Cpx

(*Evans et al., 2013*). Because A22 and mecillinam treatment decrease RcsF/BamA crosslinking, it was proposed that RcsF monitors the activity of the Bam complex (*Cho et al., 2014*). However, there is no evidence that any of the treatments that affect PG affect the function of the Bam complex. We propose an alternative explanation in which newly synthesized RcsF is engaged in signaling prior its interactions with BamA. Reduction of RcsF/OMP complexes may facilitate this type of signaling by increasing proportion of periplasmic RcsF.

Our work also provides important insights into the assembly pathway for the remarkable, inter-locked RcsF/OMP complexes. The first of these relates to BamE. *bamE* encodes one of the non-essential lipoproteins of the Bam complex (*Sklar et al., 2007*), and the function of BamE is not well-understood. Recent studies have uncovered a role for BamE in modulating BamA activity (*Rigel et al., 2012*; *2013*). However, *bamE* null mutations confer modest phenotypes with only slight defects in OMP assembly (*Sklar et al., 2007*; *Rigel et al., 2012*). Here, we show that BamE plays an essential role in assembly of RcsF/OMP complexes. In a strain lacking BamE, RcsF/OmpA complexes are almost undetectable. Nonetheless, RcsF is still recognized by BamA. In fact, increased RcsF / BamA crosslinking is observed in a *bamE* strain. This finding further supports the model that RcsF/ BamA crosslinking represents an intermediate in RcsF/OMP assembly pathway and suggests that RcsF binds BamA before it binds the OMP.

Secondly, we identified a mutation, *rcsF_A55Y* that alters a trans-lumen residue and inhibits assembly of RcsF into OMP complexes. We showed that the A55Y substitution prevents RcsF binding to BamA and this in turn prevents assembly of the RcsF/OMP complex. It is important to note that this mutation does not affect the interactions between RcsF and the downstream signaling components, as the *rscF_A55Y* mutant protein was still able to activate Rcs in response to Glb.

We do not yet understand how the inducing signal is transduced by the RcsF/OMP complexes from the cell surface to the IM components of the Rcs system. The RcsF/OMP complexes do not disassemble when OM defects are detected. Therefore, we hypothesize that small conformational changes within complex can facilitate interaction between the RcsF carboxy-terminal signaling domain and the downstream components to activate the signaling pathway and we are currently attempting to detect these changes in the RcsF/OMP complexes in response to LPS defects.

## Materials and methods

### Strain growth and construction

All strains used in this study, including previously published strains (*Majdalani et al., 2002*, *Malinverni et al., 2006*, *Malojčić et al., 2014*, *Silhavy et al., 1984*) are listed in *Supplementary file 1*. Strains were grown in LB (10 g/L tryptone, 5 g/L yeast extract, 10 g/L NaCl) at 37°C. LB was supplemented with 10 mM MgSO$_4$ when indicated. Arabinose was added at the final concertation of 0.2% to support growth of *lptE_R91D K136D* mutant. Antibiotics were added when appropriate at the following concentration: chloramphenicol 20 µg/ml, kanamycin 25 µg/ml, tetracycline 20 µg/ml. Strains were generated by P1vir transduction (*Silhavy et al., 1984*). The kanamycin resistance cassette was cured from Keio derived mutants with plasmid pCP20, as previously described (*Datsenko and Wanner, 2000*).

### NPN uptake assay

NPN uptake assay was performed according to (*Loh et al., 1984*). Briefly, AK-265 strain was grown to an OD600 of 0.5, cells were collected by centrifugation and washed twice with 5 mM HEPES, pH 7.2. Cells were resuspened to OD600=0.5 and NPN (1-N-phenylnaphthylamine, Sigma) was added to a final concentration of 10 µM. 200 µl of cell suspension was pipetted into wells of black 96 well plates. PMB was added to a final concentration of 0.5 or 8 µg/ml. Fluorescence (excitation 350 nm/ emission 420 nm) was measured for 10 min with 1 min interval using BioTek Synergy 1 plate reader. Endpoint fluorescence was normalized as a fold of untreated sample (vehicle control) and values represent mean with SD between three independent measurements.

### diSC3(5) release assay

diSC3(5) (Sigma) release assay was performed following the protocol by (*Zhang et al., 2000a*) but adopted for a plate-reader format. Briefly, AK-265 strain was grown to an OD600 of 0.5, cells were

collected by centrifugation and washed twice with 5 mM HEPES, pH 7.8. Cells were resuspened to OD600=0.05 in 5 mM HEPES, pH 7.8 with 0.2 mM EDTA. diSC3(5) was added to the final concentration of 0.4 µM. 180 µl of cell suspension was pipetted into wells of black 96 well plates. Uptake of diSC3(5) dye was monitored by a decrease in fluorescence excitation 622 nm/emission 670 nm using BioTek Synergy 1 plate reader. After fluorescence signal reached steady state (appr. 30 min), 20 µl of 1 M KCl solution was added to equilibrate the cytoplasmic and external K+ concentrations. PMB or gramicidin (Sigma) was added where applicable to the final concentration of 0.5 and 12.5 µg/ml, respectively, and incubated for 15 min. Fluorescence was measured and normalized as a fold of untreated sample (vehicle control) and values represent mean with SD between three independent measurements.

## Quantification of RNA induction by qRT-PCR

For analysis of RNA level induction of Rcs, overnight cultures of AK-265 were back diluted to $2x10^7$ cells/ml and grown to an OD600 of 0.5. Where applicable, cultures were then treated with 500 µg/ml Ksg or a vehicle control for 15 min. Cultures were then treated with 0.5 µg/ml PMB or 5 µM Glb. Samples for RNA analysis were harvested at the indicated time points and immediately fixed with a 2X volume of RNAprotect Bacterial Reagent (Qiagen, Germantown, MD) as per manufacturer instructions. The fixed cells were lysed and RNA was harvested using the RNeasy kit (Qiagen) with on column DNase (Qiagen) digestion as per manufacturer instructions. The RNA was quantitated using a Synergy H1 Hybrid Reader (BioTek, Winooski, VT) and cDNA was synthesized from 1 µg of RNA using a High Capacity cDNA reverse transcription kit (Thermo Fisher Scientific, Austin, TX) in a 20 µl reaction.

For qPCR, 10 µl reactions were run in triplicate using PerfecCTa SYBR Green FastMix R0X (Quanta Biosciences, Gaithersburg, MD), 0.5 µM primers, and 2 µl of a 1:500 dilution of cDNA samples. The following primers were utilized to quantitate *lacZ* (left: 5'-GAAAGCTGGCTACAGGAA-3'; right 5'-GCAGCAACGAGACGTCA-3'), *rpoE* (left: 5'-TGGCCTGAGCTATGAAGAGATAG-3', right: 5'CCTGATAAGCGGTTGAACTTTG-3' [*Denoncin et al., 2012*]), *cpxP* (left: 5'-TGCTGAAGTCGG TTCAGGCGATAA-3', Right: 5'-TCTGCTGACGCTGATGTTCGGTTA-3'), *nrdR* (left: 5'-ATGCA TTGCCCATTCTGTTT-3', right: 5'-CCGCTACGCAATTTCTCTTC-3'), and *ubiJ* (left: 5'-GTTATCGCC TACGCCAGTGT-3', 5'-GGCTTTGCTGATTCCTTCAG-3'). RNA expression of *rprA* was not directly analyzed as high levels of secondary structure prevent accurate quantification by qRT-PCR (data not shown). The qPCR reactions were run on the StepOne Plus RealTime PCR System (Thermo Fisher Scientific) using the StepOne Software V2.3 (Thermo Fisher Scientific) on the following program: 95°C for 10 min, followed by 40 cycles of 95°C for 15 s, 60°C for 1 min (acquisition). Absolute quantification for each primer set was performed based on Ct values calculated by automatic thresholding compared to a standard curve of $10^1$ to $10^7$ copies per reaction of *E. coli* K12 MG1655 genomic DNA. Relative expression of *lacZ, rpoE*, and *cpxP* was calculated using *nrdR* (*Figure 2A*) and *ubiJ* (*Figure 2B and C*) as endogenous control genes, as they have been found to be invariant in a wide range of conditions (*Heng et al., 2011*). For calculating fold induction, relative expression values were normalized a no treatment control for each time point. The kinetics of *lacZ* induction (*Figure 1C* upper panel) are indicated as a representative experiment with error bars representing one standard deviation. Other qRT-PCR experiments (*Figure 1D* and *Figure 2*) are indicated as the mean of three independent biological replicates+/-SEM.

## *In vivo* formaldehyde crosslinking

Strains were grown to an OD600 of 0.5–0.7, washed twice in PBS ($Na_2HPO_4$ 10mM, $KH_2PO_4$ 1.8 mM, KCl 2.7 mM, NaCl 137 mM, pH 7.6) and concentrated to OD600=10 in PBS. Formaldehyde was added to the final concentration of 0.7% and crosslinked for 12 min at room temperature. Crosslinking was stopped by adding of Tris-Cl (100 mM final); cells were collected by centrifugation and suspended in SDS loading buffer. Samples were heated at 65° C for 15 min and analyzed by immune blotting with anti-RcsF antibodies.

## β-galactosidase assay

For a time course experiments, overnight cultures of corresponding strains were diluted 1:100 in LB (supplemented with kanamycin 25 µg/ml when needed) and grown to an OD600 of 0.5 in a shaking

water bath at 37°C. Cultures were then treated with 0.5 µg/ml PMB or 5 µM Glb for a course of 60 min. Samples were taken every 10 min for a β-galactosidase assay and for OD600 measurement. For a β-galactosidase assay, 100 µl samples were taken and added directly to 900 µl of Z buffer (60 mM $Na_2HPO_4$, 40 mM $NaH_2PO_4$, 10 mM KCl, 1 mM $MgSO_4$, 50 mM β-mercaptoethanol, 0.03% SDS). 50 µl of chloroform was added to stop growth and mix vigorously by pipetting. After collecting all samples, 100 µl of cell each lysate was mixed with 100 µl of 4 mg/ml O-nitrophenyl-β-D-galactopyrano-side (ONPG) solution in Z buffer. β-galactosidase activity was analyzed by a kinetic measurement of OD420 in a BioTek Synergy 1 plate reader and Vmax was determined using Gen5 software. Vmax was normalized by OD600. Experiments were performed in three biological replicates and mean values +/- SEM were plotted. Graphs were built by GraphPad Prism 6 software. For a single point measurement, corresponding strains were grown to an OD600 of 0.5–0.7 and samples for a β-galactosidase assay were taken and analyzed as described above.

## Acknowledgements

We thank members of the Silhavy lab for helpful discussions and the feedback on the manuscript. This work was supported by National Institute of General Medical Sciences Grant GM34821 (to TJS).

## Additional information

### Funding

| Funder | Grant reference number | Author |
| --- | --- | --- |
| National Institute of General Medical Sciences | GM34821 | Thomas J Silhavy |

The funders had no role in study design, data collection and interpretation, or the decision to submit the work for publication.

### Author contributions

AK, AMM, Conception and design, Acquisition of data, Analysis and interpretation of data, Drafting or revising the article; TJS, Conception and design, Analysis and interpretation of data, Drafting or revising the article

### Author ORCIDs

Anna Konovalova, http://orcid.org/0000-0002-2238-8849
Thomas J Silhavy, http://orcid.org/0000-0001-7672-5153

## Additional files

### Supplementary files

• Supplementary file 1. Strains used in this study.

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
