## [Decision Letter]

Thank you for submitting your article "A lipoprotein/β-barrel complex monitors lipopolysaccharide integrity transducing information across the outer membrane" for consideration by *eLife*. Your article has been reviewed by two peer reviewers, and the evaluation has been overseen by a Reviewing Editor and Michael Marletta as the Senior Editor. The following individuals involved in review of your submission have agreed to reveal their identity: Timothy Meredith (Reviewer #1).

The reviewers have discussed the reviews with one another and the Reviewing Editor has drafted this decision to help you prepare a revised submission.

Your previous findings showed that RcsF forms a complex in which it is threaded through the lumen of outer membrane proteins such as OmpA. In this paper you report that the RcsF/OMP complexes can induce Rcs system through OM stress response by showing that pre-synthesized RcsF is sufficient to induce Rcs in the absence of de novo protein synthesis, as shown using chemical inhibitors. This challenges the model of Choi et al. for RcsF induction, which evokes a requirement for de novo protein synthesis. Clearly, the model is evolving and your manuscript is an important step towards this. Overall, the conclusions are well supported and provide new insight into complex signaling events. The main concern is whether it has been proven that changes in LPS structure are directly detected by RcsF, or if the sequence of events is LPS mutation leading to "altered OM", which in turn activates RcsF signaling. A few suggested control experiments should help to strengthen your conclusions that it is indeed the former scenario, as you have proposed.

Essential revisions:

1) Introduction section, paragraph four: Is the RcsF/IgaA interaction the only known partner for RcsF?

2) Results section, paragraph two: Why was the lipid truncated polymyxin analog nonapeptide not used? This analog avoids cell death and toxicity yet still can bind to and disrupt LPS interactions.

3) Subsection “RcsF/OMP complexes are required for sensing OM stress”, paragraph two: "PMB did not induce Rcs in the ompA mutant"- was the MIC to PMB tested in this strain background? If the MIC were to change or become higher in ompA mutant, this could potentially cause the apparent uncoupling of Rcs induction from PMB. At minimal, PMB MIC should be measured and reported for this mutant in particular.

4) Same section, paragraph three: Protein levels may not change, but what is known about the function/phenotype of complexed OmpA/RcsF verses free OmpA (as in mutants defective in complex assembly)? e.g. can more uncomplexed OmpA influence some of the reported phenotypes?

5) Same section, final paragraph: This argument is not fully convincing. In Figure 3, the A55Y mutant shows significant induction in the waaP deletion, comparing it to its waaP+ parent (it would be useful to have the actual numbers to compare in some of these cases), suggesting that this strain still responds (as well as WT?), but starts at a lower level (additive effects, rather than any epistasis?). It might be useful to discuss why this derivative acts like a null in the waaP+ strain. For bamE, the argument is stronger, since the waaP derivative is similar to WT and to bamE waaP+. The authors need to discuss this result; does A55Y still make complexes with other OMPs that are stimulatory in the waaP context? Or does another pathway become activated?

6) Subsection “Positive charge of the surface-exposed region of RcsF is required for LPS-sensing”: More explanation is needed to document how you conclude that decreased phosphorylation of LPS is sufficient. The statement in the Discussion seems like an overstatement – the results are consistent with this model, but what really shows it?

7) Same section, paragraph two: What are the permeability characteristics of the WaaC, WaaF, and WaaG mutations? Inner core phosphorylation mutations can have significant affects on permeability (as measured by MIC to hydrophobic compounds), which would complicate conclusions that Rcs is induced through altered LPS structures.

8) Paragraph two, same section: If we understand correctly, the conclusion, which states that RcsF senses altered LPS structures rather than OM permeability, it appears to be inconsistent with conclusions, which report Mg^2+^ ion rescues through stabilizing LPS lateral interactions. Since Mg^2+^ does not change the structure of LPS, wouldn't this suggest that it could be through stabilization of OM?

9) Paragraph there, same section: The composition of the basal media here should be stated (i.e. was it normal LB, which has sufficient Mg/Ca to keep PhoPQ mostly repressed).

10) Paragraph four, same section: Is Rcs significantly activated in these mutants so as to induce mucoidy? Would this be expected based on the waaP mutation? Are any of the deep rough mutants tested mucoid?

11) Figure 3: Given that multicopy rcsF is known to induce the rcs system on its own, it would be good to include evidence here or in a supplemental figure that this rcsF plasmid does not induce without treatment. In addition, it would be easier for the reader if the two graphs in 3C were labeled with the PMB or GLB treatment (also true for 5C).

12) Paragraph four, same section, Figure 5: An issue similar to that in point 5 arises with Figure 5 (the A5 mutant is significantly increased in the waaP strain background). Therefore, the conclusion that the positive charge of the surface-exposed region is critical is not fully convincing. Is there evidence that the structure of RcsF protein (with the N-terminus of the protein on the outside of the cell) is preserved in this mutant?

13) Figure 4: Would polymyxin or the waaP mutant have any effect in these mutants? One such experiment is important to confirm that the permeability, etc., doesn't actually block the response (rather than simply not inducing it).

14) Given the effects of components of the phoP/Q regulon on LPS, what does full-blown induction of phoP/Q (is there a constitutive mutant, to avoid Mg effects) do to RcsF sensing? Are these set up to be either/or signaling pathways, or do they collaborate/reinforce each other.

---

## [Author Response]

1) Introduction section, paragraph four: Is the RcsF/IgaA interaction the only known partner for RcsF?

IgaA is the only known partner among the downstream signaling components. RcsF interactions with OMPs and BamA are discussed separately in the context of topology and assembly.

2) Results section, paragraph two: Why was the lipid truncated polymyxin analog nonapeptide not used? This analog avoids cell death and toxicity yet still can bind to and disrupt LPS interactions.

Initially, we did not use polymyxin nonapeptide (PMBN) because it is not bactericidal and does not have an MIC. Therefore, it is hard to estimate the minimal amount of peptide to use without causing secondary effects. However, we do agree that Rcs induction by PMBN would provide additional support our models, in which PMB induces Rcs by disrupting lateral interactions due to the charge neutralization. PMBN has been previously shown to induce Rcs in *Salmonella* (Farris, 2010). We now also show concentration-dependent induction of Rcs by PMBN in *E. coli* (Figure 4). PMBN induction does require increased concentrations to offset a well-documented decrease in the binding affinity of PMBN to LPS.

*3) Subsection “RcsF/OMP complexes are required for sensing OM stress”, paragraph two: "PMB did not induce Rcs in the ompA mutant"- was the MIC to PMB tested in this strain background? If the MIC were to change or become higher in ompA mutant, this could potentially cause the apparent uncoupling of Rcs induction from PMB. At minimal, PMB MIC should be measured and reported for this mutant in particular.*

We include the MIC values as a Figure 3—figure supplement 1 to show that the *ompA* as well as the *bamE* mutant does not have increased or decreased sensitivity to PMB. More importantly, we now include corresponding growth (OD600) plots for all of our time course experiment to show that 1) The PMB concentration used does not affect growth, and 2) none of the strains had an altered growth phenotype under conditions of PMB treatment (Figure 3—figure supplement 1).

*4) Same section, paragraph three: Protein levels may not change, but what is known about the function/phenotype of complexed OmpA/RcsF verses free OmpA (as in mutants defective in complex assembly)? e.g. can more uncomplexed OmpA influence some of the reported phenotypes?*

OmpA is one of the most abundant proteins in *E. coli* and exists in at least 100x excess of RcsF. Considering that RcsF interacts with multiple proteins, we estimate that less than 0.5% of OmpA exists in complex with RcsF. Therefore, we do not believe that increased concentration of OmpA is responsible for any of the reporter phenotypes.

*5) Same section, final paragraph: This argument is not fully convincing. In Figure 3, the A55Y mutant shows significant induction in the waaP deletion, comparing it to its waaP+ parent (it would be useful to have the actual numbers to compare in some of these cases), suggesting that this strain still responds (as well as WT?), but starts at a lower level (additive effects, rather than any epistasis?). It might be useful to discuss why this derivative acts like a null in the waaP+ strain. For bamE, the argument is stronger, since the waaP derivative is similar to WT and to bamE waaP+. The authors need to discuss this result; does A55Y still make complexes with other OMPs that are stimulatory in the waaP context? Or does another pathway become activated?*

Figure 3 shows that although RcsF/OmpA crosslinking is significantly reduced in *rcsF_A55Y*, it is not completely abolished. Therefore, we attribute some low level of Rcs induction in the *waaP* strain to the residual RcsF/OMP complexes (we added this discussion to the text). This level of induction, however, is not sufficient to complement the *rcsF waaP* mutant for both *PrprA-lacZ* expression (Figure 3) as well as expression of capsule genes, and we now include a figure supplement xx to show that the *waaP* strain with *rcsF_A55Y* mutation is not mucoid (Figure 3—figure supplement 2), similar to *bamE waaP*.

*6) Subsection “Positive charge of the surface-exposed region of RcsF is required for LPS-sensing”: More explanation is needed to document how you conclude that decreased phosphorylation of LPS is sufficient. The statement in the discussion seems like an overstatement – the results are consistent with this model, but what really shows it?*

*waaCFG* mutants have a truncated core and decreased phosphorylation. The *waaP* mutant produces full-length LPS but has decreased phosphorylation. Because *waaP* induced Rcs to the same extend as *waaCFG*, we concluded that decrease of phosphorylation is sufficient for induction. We added more explanation in the text.

*7) Same section, paragraph two: What are the permeability characteristics of the WaaC, WaaF, and WaaG mutations? Inner core phosphorylation mutations can have significant affects on permeability (as measured by MIC to hydrophobic compounds), which would complicate conclusions that Rcs is induced through altered LPS structures.*

Inner core mutations are known to affect permeability and we acknowledged this in our paper. This is why we showed that Rcs was not induced in *lptD4213, mlaA pldA* and *lptE613* strains before we did detailed characterization of the *waa* mutants to rule out indirect effects. The following changes were made to clarify this question:

1- we changed the order of the results presented, starting with a genetic analysis of Rcs induction in *waa* mutants, and then followed by showing that mutants with severe OM permeability do not induce Rcs. The permeability defects caused by *lptD4213, mlaA pldA* and *lptE613* are known and are similar to those of the LPS inner core biosynthesis/phosphorylation mutants.

2-To make our argument more visual, we also show that *lptD4213, mlaA pldA* and *lptE613* mutants also display OM permeability defects. The sensitivity of *lptD4213, mlaA pldA* mutants to detergents and rifampicin, which are the characteristics of *waa* mutants, are documented by including the efficiency of plating assays (Figure 4). We also added the original references describing the permeability phenotype of these mutants in great detail.

8) Paragraph two, same section: If we understand correctly, the conclusion, which states that RcsF senses altered LPS structures rather than OM permeability, it appears to be inconsistent with conclusions, which report Mg^2+^ ion rescues through stabilizing LPS lateral interactions. Since Mg^2+^ does not change the structure of LPS, wouldn't this suggest that it could be through stabilization of OM?

We clarified that RcsF senses LPS lateral interactions which can be disrupted either by LPS charge neutralization due to binding of PMB, or genetically by removing the majority of the phosphates. Mg^2+^ mediates LPS lateral interactions; LB media is a poor source of Mg^2+^ and the amounts present are not sufficient to saturate all of the LPS molecules (Papp-Wallace and Maguire 2008; Nikaido 2009). It is true that Mg^2+^ does not alter LPS structure (just like PMB does not), but the lack of Mg^2+^ also leads to destabilized lateral interactions. The *waaP* mutant still contains lipid A phosphates (an essential LPS modification in *E. coli*). Addition of Mg^2+^ would help to stabilize lateral interactions in this strain through the lipid A phosphates. We included this discussion in the text.

9) Paragraph there, same section: The composition of the basal media here should be stated (i.e. was it normal LB, which has sufficient Mg/Ca to keep PhoPQ mostly repressed).

We added LB composition and the amount of Mg^2+^ used in Materials and methods.

*10) Paragraph four, same section: Is Rcs significantly activated in these mutants so as to induce mucoidy? Would this be expected based on the waaP mutation? Are any of the deep rough mutants tested mucoid?*

The mucoid phenotype of these mutants is well known and is one of the characteristics of deep rough mutants. We included the description of the mucoid phenotype of *waaC, F, G* and *P* mutants alongside references.

11) Figure 3: Given that multicopy rcsF is known to induce the rcs system on its own, it would be good to include evidence here or in a supplemental figure that this rcsF plasmid does not induce without treatment. In addition, it would be easier for the reader if the two graphs in 3C were labeled with the PMB or GLB treatment (also true for 5C).

We used a low copy *rcsF* plasmid and we show in Figure 3 and Figure 5 that it does not confer full induction of Rcs like *rcsF* overexpression plasmids do. We now include untreated curves for all of the strains used in the study as Figure 3—figure supplement 1 and Figure 5—figure supplement 1.

*12) Paragraph four, same section, Figure 5: An issue similar to that in point 5 arises with Figure 5 (the A5 mutant is significantly increased in the waaP strain background). Therefore, the conclusion that the positive charge of the surface-exposed region is critical is not fully convincing. Is there evidence that the structure of RcsF protein (with the N-terminus of the protein on the outside of the cell) is preserved in this mutant?*

We state clearly in the paper that positive charge is not the only contributing factor to RcsF activity. The *rcsF_A5* mutant is significantly affected in PMB sensing and in activity in the *waaP* background. However, like in the case of A55Y, it cannot fully complement the *rcsF waaP* mutant for both *PrprA-lacZ* expression (Figure 5) and expression of capsule genes. To highlight this decreased function, we now include a Figure supplement to show that the *waaP rcsF_A5* strain is not mucoid (Figure 5—figure supplement 2).

RcsF surface exposure depends on RcsF assembly into RcsF/OMP complexes and this is a more quantitative phenotype than are the dot blot assays. RcsF_A5 is assembled into RcsF/OMPA complexes as efficiently as the WT (Figure 5), demonstating that rcsF_A5 adopts the same topology as the WT.

The surface exposed region of RcsF is unstructured because it contains 25% proline residues distributed throughout the region (Figure 5). Because it is so proline-rich it likely remains as unstructured in the A5 mutant as it is in the WT. In addition, because RcsF_A5 is assembled properly (Figure 5) into the heterodimeric complexes and is functional for Glb signalling (Figure 5), there is no reason to believe that a critical structure of RcsF_A5 is affected in any way.

*13) Figure 4: Would polymyxin or the waaP mutant have any effect in these mutants? One such experiment is important to confirm that the permeability, etc., doesn't actually block the response (rather than simply not inducing it).*

RcsF signalling is not generally inhibited in *lptD4213, mlaA pldA* and *lptE613* mutants because it can be stimulated by the *waaP* mutation. We now include this data in a Figure 4.

*14) Given the effects of components of the phoP/Q regulon on LPS, what does full-blown induction of phoP/Q (is there a constitutive mutant, to avoid Mg effects) do to RcsF sensing? Are these set up to be either/or signaling pathways, or do they collaborate/reinforce each other.*

The reviewers refer to the to the *phoPQ* effect of LPS modification in *Salmonella* through Mmr pathway. PhoPQ in *E. coli* does not regulate these genes and does not control LPS modifications, and hence has no effect on LPS structure (Winfield and Groisman 2004; Rubin et al. 2015). We include this statement in the text to avoid confusion. Null mutations in *phoPQ* are universally used to test the contribution of *phoPQ* mutations to the phenotype. We clearly showed that the *phoP* null mutant of *E. coli* still displays Mg^2+^-dependent Rcs induction in the *waaP* background and that PhoPQ is not involved in this signalling.

The Rcs/Pho regulons have overlap in different enterobacteria and have been the focus of the work of others, (citations included in the text). However, Rcs and especially the Pho regulons vary greatly from species to species (e.g. the Pho regulon is completely different in *E. coli* compared to *Salmonella,* (Monsieurs et.al. 2005), we don’t feel confident to broadly discuss the functional significance of such overlap.